# Real-Time Monitoring of Self-Fed Supplement Intake, Feeding Behaviour, and Growth Rate as Affected by Forage Quantity and Quality of Rotationally Grazed Beef Cattle

**DOI:** 10.3390/ani9121129

**Published:** 2019-12-12

**Authors:** José A. Imaz, Sergio García, Luciano A. González

**Affiliations:** 1School of Life and Environmental Sciences, Faculty of Science, The University of Sydney, Sydney, NSW 2570, Australia; sergio.garcia@sydney.edu.au (S.G.); luciano.gonzalez@sydney.edu.au (L.A.G.); 2Instituto Nacional de Tecnología Agropecuaria (INTA), Capital Federal 1033, Argentina; 3Sydney Institute of Agriculture, The University of Sydney, Sydney, NSW 2570, Australia

**Keywords:** technologies, data, supplementation, forage quantity, feeding behaviour, management

## Abstract

**Simple Summary:**

In grazing systems, the use of novel technologies such as electronic feeders and automatic weighing systems enables collection of daily data of cattle feeding behaviour and growth. These technologies can be useful to study animal response to varying forage quantity and quality within and throughout grazing periods and seasons. The aim of this 254-day experiment was to investigate the effect of forage type, quantity, and quality on the consumption of a self-fed supplement (molasses-lick blocks (MLB)) and on the growth rate and feeding behaviour of grazing beef cattle. Results indicated that type and amount of forage affect MLB intake and feeding behaviour. Thus, when feed availability is low (e.g., forage on the paddock is depleted), animals increase consumption of and number and duration of visits to the supplementary feed. The use of MLB only improved growth rate of cattle when animals grazed sorghum and pasture or were fed oaten hay. Monitoring the feeding behaviour of animals around MLB reflects changes in forage quantity and quality.

**Abstract:**

Supplement intake and liveweight (LW) data were collected daily and remotely by digital in-paddock technologies (electronic feeder (EF) and walk-over-weighing scale (WOW)) to study the effect of forage quantity and quality on the intake of a self-fed supplement (molasses-lick blocks (MLB)), LW, liveweight change (LWC), and feeding behaviour of grazing beef cattle. Fifty-two crossbred weaners were rotationally grazed or fed for 254 days on different forages: sudangrass (SG), autumn pastures (P), winter pastures with concentrate (P+C), oat crops (OC), lucerne hay (LH), and oaten hay (OH). Forage quantity and quality were measured on the day of entry (high feed availability) and exit (low feed availability) stages of grazing or hay delivery. The intake of MLB was 111% higher (*p* < 0.05) at low compared to high feed availability, and this was also reflected in the feeding behaviour of animals (e.g., greater feeding frequency and rate). Moreover, there was a large temporal variability of daily MLB intake (Coefficient of variation (CV) = 146.41%). Supplementing MLB improved LWC only with SG, P, or OH (*p* < 0.05). The behaviour of animals around MLB reflects changes in feed quantity and quality and could be used to enhance cattle grazing and nutritional management in real time.

## 1. Introduction

Efficient and profitable pasture-based beef production systems rely on sustained animal growth rates over entire seasons. Animals are typically supplemented to cover true pasture deficits or to increase performance, but it is difficult to monitor temporal changes of supplement intake and liveweight (LW) of grazing cattle without human intervention in near real time [1]. However, recent technological advances like electronic feeders and in-paddock walk-over weighing scales (WOW) allow producers to overcome this limitation by monitoring individual animals and/or whole herd performance [2].

Electronic feeders can be also used to assess the feeding behaviour of animals (i.e., frequency and duration of single feeding events) associated with variations in supplement intake through different types of feedstuffs [3]. These technologies can be particularly important under grazing conditions, where complex interactions between forage quantity, quality, and characteristics of supplement drive intake and growth responses [4]. Additionally, feeding behaviour could be used to predict supplement intake of grazing animals. 

Simultaneous and frequent (e.g., daily) collection of data on supplement intake and LW can be useful for beef producers who have cattle grazing on different types of forages through the seasons, with the associated variability in liveweight gain (growth rate). Assessing cattle liveweight change (LWC) frequently (e.g., daily or weekly) and in real time could help with early interventions when changes in growth are detected. Timely management would allow producers to graze paddocks to optimize LWC, to introduce feed supplementation, and to adjust stocking rate. The WOW could also include gates for auto-drafting animals into different pens or yards, which would allow more precise feed supplementation of groups of animals for a target animal performance.

In practice, most supplements cannot be offered as “free-choice” or ad-lib due to animal health and economic reasons. However, molasses-lick blocks (MLB) are self-fed supplements which aim to control MLB intake through the block hardness [5] while offering protein, minerals, and feed additives (e.g., ionophores, urea, and oil). Previous research suggests that MLB intake and feeding behaviour of cattle can be affected by several factors, including forage quantity and quality and the availability of other feed supplements [6,7]. However, the effects of these factors have not been investigated measuring MLB intake and LWC of a group of animals with concurrent measurements of feed quantity and quality [8]. It is also known that molasses-based supplements could have positive impacts on LWC when consuming low-quality forages, potentially due to the improvements on feed digestion and dry matter intake [9,10]. However, none of these experiments measured animal responses frequently (i.e., daily or weekly LW) for a long period of time (i.e., throughout seasons) as forage quantity and quality change.

Therefore, the objectives of the present study were (a) to monitor the effects of MLB supplementation on cattle LW under grazing conditions using electronic feeder (EF) and WOW and (b) to assess the dynamic relationship between MLB intake; feeding behaviour; and forage type, quantity, and quality. We hypothesized that the intake of MLB and feeding behaviour of cattle was affected by changes in forage quantity and quality.

## 2. Materials and Methods

The experiment was conducted at John Pye Farm (latitude: 33°56′93″ S, longitude: 150°40′47″ E, Greendale, NSW, The University of Sydney). All experimental procedures were approved by The University of Sydney Animal Ethics Committee (Approval 2017/1162).

### 2.1. Experimental Details

Fifty-Two Charolais × Angus crossbred weaners (mean ± SD, initial LW = 177.95 ± 31 kg/hd; initial age = 187 ± 43 days) were tagged with electronic identification (EID), blocked by sex and LW, and randomly assigned to one of two treatments: (a) No MLB supplementation (NS) and (b) access to a single MLB (40 kg; 4 Season Co. Pty Ltd, Creastmead, Queensland, Australia) made available inside an EF throughout the experiment as free choice. The chemical composition of the MLB was [Dry Matter (DM) basis] Crude Protein (CP) = 8.9%, Neutral Detergent Fiber (NDF) = 2.83%, and Dry Organic Matter Digestibility (DOMD) = 71.5%, and the ingredient composition was 42% molasses, 9% salt, 3% urea, 3% vegetable oil, 1.3% phosphorus, 3% calcium, 4% magnesium, 15% cottonseed (*Gossypium hirsutum*) meal, 2% Lasalocid (Bovatec, Zoetis, Parsippany, New Jersey), 6% trace mineral mix (copper, cobalt, iodine, and zinc), and 11.7% water. A total area of 24.7 ha of pastures and annual crops was rotationally grazed for 254 days. The paddocks with annual crops (6.7 ha) were subdivided into cells of approximately 1 ha using electric fences. Total area with mixed pastures (18 ha) was subdivided into 9 paddocks, which ranged from 0.8 to 6.65 ha. The average grazing time on each paddock during the trial was 10.1 ± 5.3 days. Additionally, hay supplementation was introduced from 12 September to 22 November 2017. Total rainfall during the experiment was 239 mm, and monthly rainfall was 68.8, 10.0, 12.7, 58.0, 0.0, 22.4, 0.0, 53.7, and 13.0 from March to November, respectively. No irrigation was applied. Average stocking rate was 2.5 hd/ha ranging from 13.7 dry sheep equivalents (DSE)/ha to 23.2 DSE/ha.

Animals were grazed on temperate pastures and annual crops and were supplemented with hay and concentrate. These feed types were (a) Sudangrass (SG; *Sorghum vulgare* var. sudanense) grazed from day 1 (when animals were moved to the final trial location) to day 26; (b) autumn pastures (P) grazed from day 27 to 121; (c) oat crops (OC, *Avena sativa*) grazed from day 122 to 156; (d) winter pastures with concentrate supplementation (P+C) from day 157 to 185; (e) Lucerne hay (*Medicago sativa*, LH) offered from day 186 to 220; and (f) oaten hay (*Avena sativa*, OH) offered from day 221 to 254. It is important to stress that MLB-supplemented and NS animals co-grazed the paddocks and were sorted automatically into treatments at the entrance of the yard every time they wanted to access water (Figure 1).

Feed availability was classified as early (high feed availability) or late (low feed availability) during a paddock grazing which coincided with the highest or lowest forage availability per ha, respectively, or with the day when hay was delivered or not. Molasses-lick-block intake and feeding behaviour on high and low feed availability from P, OC, and P+C (for which grazing time per cell was greater than 10 days) was estimated by averaging as follows: (a) The first 2 days after entering to a fresh paddock (feed availability = high); (b) the last 2 days before leaving a paddock (feed availability = low). When paddock utilisation was shorter than 7 days, only the first and the last days were considered as high and low, respectively. Molasses-lick-block data from LH and OH supplementation was pooled from 14:00 h of the “feed delivery day” to 14:00 h of the next day (24 h period, feed availability = high). The remaining days were considered “no feeding day” until a new hay bale was delivered (14:00 h of the current day to 14:00 h of the next day; feed availability = low).

### 2.2. Walk-Over-Weighing and Electronic Feeder Setup

A WOW with an auto drafter gate (Precision Pastoral Ltd, Alice Spring, Northern Territory, Australia) and an EF (Smartfeed developed by C-lock Inc., Rapid City, SD, USA) were installed at the only central water point. The WOW recorded animal EID, date, time, and LW whereas the EF recorded EID, time, date, and the weight of feed disappearing in each visit to the feeder. A description of dimensions and operation of each technology was reported by González et al. [2] and Reuter et al. [11] for WOW and EF, respectively.

A yard (15 m × 25 m) was built at a central location of the paddocks grazed in the experiment and subdivided into two equal-sized sections, each of them sharing a single water point (Figure 1). The yard had a single entry where the WOW was located, and each section of the yard had a single exit (Figure 1, numbers 5 and 6). Spear gates were used at the entry to the WOW and each exit, allowing animals to move in only one direction. The auto-drafter was placed immediately after the weighing station to allow animals to be automatically drafted into a supplemented (left) and not-supplemented (right) group. Additionally, the auto-drafter can enable all cattle to remain together as a group in the same paddock, offering equal grazing conditions. An EF was placed in the yard section of the supplemented animals on 10 April (day 28) aside the exit gate (Figure 1) and firmly anchored to the ground. Feeding events started to be recorded on 16 of April (day 34).

Animals were gradually trained to use the WOW system and EF. During the first 4 weeks from weaning to the beginning of the experiment, animals could recognize and walk through a WOW following the training procedure proposed by Gonzalez et al. [2]. Finally, animals were moved to the final trial location and randomly allocated to one of two treatments described above. The EF contained a pneumatic gate, which was left continuously open for 2 weeks, so the animals could recognize the feed bin as the feed container. At week 3, the pneumatic gate was set to close halfway (50% closed) and to open when an animal approached it to eat. Finally, the feeder gate was set to allow full closing on week 4. Feeding behavior and MLB intake were recorded from the first day after installation. Animal attendance to the water point, the auto-drafter operation, and feeder usage were monitored daily, analysing the list of animals that have been read by both the WOW and EF. During the first 6 weeks of the experiment, supplemented animals had one available MLB outside the EF, placed in a central location of their yard, to expose all animals to the supplement. Disappearance of the MLB placed outside was recorded manually by weighing the block three times per week (Monday, Wednesday, and Friday at 10:00 a.m.).

### 2.3. Grazing and Supplementation Management and Measurements

Concentrate and hay supplementation were introduced due to reduced grass growth during the winter and due to drought and were delivered infrequently on Monday, Wednesday, and Friday. Pellets (CP: 16.2%; NDF: 35.3%; DOMD; 68.7%) and chopped lucerne-chaff (CP: 15.8%; NDF: 41.1%; DOMD: 54%), mixed in a proportion of 75:25, were offered from 07 August (day 147) to 12 September (day 183) at a rate of 1.25 kg as fed/hd per day. While feeding hay (periods E and F), animals also had a pasture paddock available under continuous grazing. Pre-grazing forage quantities were 1130 and 830 kg DM/ha to a base of 5 cm at the beginning of periods E and F, respectively. During these periods E and F, square bales were weighed prior to delivery (375 kg as fed for LH and 425 kg as fed for OH) and its availability was monitored daily by placing automatic camera traps taking a single photo of the hay every five minutes. Hay wastage was estimated on six bales per feed type by weighing the remaining hay immediately before offering a new bale and by correcting by the DM content of the collected material. The estimated values of hay utilization efficiency were 94 and 78% for LH and OH, respectively. Additionally, two samples of 300 g of hay were taken on each feeding day and a weekly composite sample was obtained for chemical analysis after mixing.

The pasture mix contained perennial ryegrass (*Lolium perenne*), fescue (*Festuca arundinacea*), white clover (*Trifolium repens*), Cocksfoot (*Dactylis glomerata*) and Chicory (*Chicorium intybus*), Kangaroo grass (*Themeda triandra Forsk* syn *australis*), Paspalum (*Paspalum dilatatum Poir*.), Purple pigeon grass (*Setaria incrassate* cv. Inverell), Setaria (*Setaria sphacelata* var. seric), and Rhodes grass (*Chloris gayana Kunth*). Predominant species were Fescue, Cocksfoot, Paspalum, and Rhodes grass. Annual crops (SG and OC) were double cropped on the same location; previous fertilization was with 50 kg/ha of nitrogen, 22 kg/ha of phosphorus, and 50 kg/ha of potassium before sowing for each crop. Animals were moved to a fresh paddock when forage quantities to the base of 5 cm were approximately 1500, 1000, and 750 kg DM/ha for SG, pastures, and OC, respectively. For SG, ten 0.5 m^2^ quadrats were cut manually before and after each paddock was grazed. For pastures and OC, forage quantity was measured using an electronic plate meter (EC20, NZ Agriworks Ltd, Feilding, New Zealand). For this purpose, a calibration model between forage quantity (kg/DM per ha) and pasture height measured with the plate was done by manually cutting to the base of 5 cm fifty 0.25 m^2^ quadrats for each forage type. Linear models were obtained and used to estimate forage DM quantity (R^2^ = 0.73 for pastures and R^2^ = 0.83 for OC). Forage sampling for chemical composition consisted of six cut samples taken to the base of 5 cm with a 0.25 m^2^ quadrat before and after grazing of each paddock. The first two samples were used to determine DM content (%) and the next two samples were used to analyse chemical composition. The last two samples were used to determine the proportion of green and dead material by separating both fractions manually prior to drying.

### 2.4. Chemical Analysis

The DM of forage samples were determined by drying at 60 °C for 72 h in a forced-air oven (Heraeus, D-63450 Hanau). Samples for chemical analysis were dried following the same procedure and ground through a 1-mm sieve screen prior to analysis (Restsch, SM 100). Chemical composition [NDF, Acid Detergent Fiber (ADF)], CP, ash content, organic matter (OM), dry matter digestibility (DMD), and DOMD) of forage samples were estimated using near-infrared spectroscopy (The Feed Quality Service, Wagga Wagga Agricultural Institute, Department of Primary Industries, Wagga Wagga, Australia) using a Bruker MPA FT-NIR instrument in conjunction with OPUS ver. 7.5.18 (Bruker Optik GmbH, Ettlingen, Germany). Samples obtained from MLB and pellets were analyzed for NDF and ADF in accordance with Reference [12] using an Ankom 220 model fiber analyzer (ANKOM, Technology Macedon, NY, USA). Nitrogen concentration was determined using the DUMAS method according to AOAC Crude Protein 990.03 on a LECO Trumac combustion Analyzer (LECO Corporation, Saint Joseph, Michigan, USA) and using a factor of 6.25 for conversion to CP. Dry matter digestibility was analysed using the Pepsin–Cellulase Method [13].

### 2.5. Statistical Analysis

The statistical analysis of forage data from grazed paddocks was performed using a mixed-effects linear regression model with paddock as random effect and feed type and feed availability 2-way interactions as fixed effects. Forage data during hay feeding were analysed separately using a mixed-effects linear regression model with feed type as fixed effect and week of sampling as repeated measures.

Daily MLB intake and feeding behaviour data from individual animals were averaged by feed type (P, P+C, OC, LH, and OH) and feed availability (high and low) and then analysed using mixed-effects linear regression models including feed type, feed availability, sex, and 2- and 3-way interactions as fixed effects with animal as random factor. Daily MLB intake, feeding frequency, and feeding duration data were transformed to log_10_ prior to analysis.

Liveweight data were analysed following the procedure proposed by Gonzalez et al. [2], which consists of deleting records containing missing EID, extreme weights, and fitting data to penalised B-Splines for each individual animal. Animal growth rate (LWC, kg/hd per day) was calculated as the first derivative of the predicted LW curve, and the resulting LW data were averaged by date for each animal if more than one measurement per day and animal existed. The statistical analysis of LW and LWC was done using linear mixed-effects models where MLB supplementation, sex, date, and 3-way interactions were fixed effects and animal was a random effect.

Data from EF were analysed following the next steps: (1) Records without EID were deleted (n = 457, 12%); (2) records with negative feed intake were deleted (n = 652, 17%); (3) a correlation model between time (sec) and MLB intake (g) was fitted, and residuals were calculated; and (4) records with residuals lower or higher than 3 were deleted (n = 20 adding up to 69.52 kg of MLB). Feeding records in the final dataset had a mean ± SD of 0.169 ± 0.213 kg, whereas deleted feeding events had 3.47 ± 2.17 kg. Resulting MLB intake data (n = 2661 visits) were summed by date for each animal (g/hd per day). Also, daily feeding frequency (visits/hd per day), feeding duration (min/hd per day), visit length (min/visit), and visit size (g/visit) were calculated. Feeding rate was calculated for each day dividing MLB intake (g/hd per day) by total duration (min/hd per day). Daily MLB intake was transformed to log_10_ and analyzed using mixed-effects linear regression models including date, sex, and 2-way interactions as fixed effects with date as the repeated factor for each animal. Covariance structure was selected based on the lower Bayesian criterion (BIC). Least square means were calculated, and differences between means were corrected for multiple comparisons using Bonferroni test. Statistical significance was declared at *p* < 0.05. Coefficient of variation (CV) of daily MLB intake was calculated using values obtained from the linear model. All linear model procedures were done using SAS Software (SAS Institute Inc., Cary, NC, USA).

## 3. Results

### 3.1. Feed Quantity and Quality 

Forage quantity and quality differed across feed types as cattle grazed different pastures throughout the seasons (*p* < 0.05; Figure 2 and Table 1). However, the interaction between feed type and feed availability (*p* < 0.001; Figure 2a) indicated that the difference between pre- and post-grazing was larger for SG compared to P and OC (Figure 2, *p* < 0.05). Oats winter crop was grazed in an early vegetative stage, resulting in the highest proportion of green forage quantity both pre- and post-grazing (*p* < 0.01). Pre- and post-grazing forage quantity did not differ for P+C (*p* > 0.05). In contrast to forage availability, no feed type × feed availability interaction (*p* > 0.05) was observed for CP, NDF, ADF, DMD, and DOMD during these grazing periods (Table 1). However, forage quality was lower post- compared to pre-grazing for all feed types (*p* < 0.05; Table 1). Oaten hay had higher forage quantity (*p* < 0.05; Figure 2b) and lower quality (CP, NDF, ADF, and DMD) compared to LH (*p* < 0.05; Table 2).

### 3.2. MLB Supplement Intake

Herd average daily MLB intake was affected by date and sex (*p* < 0.05), but their interaction (*p* > 0.10) ranged on average from 0 g/hd per day during the pasture period to 705.50 while on OH, with a large variability through time (CV = 146.41%; Figure 3a). Average of MLB intake over the entire period was 74.04 ± 17.40 g/hd per day being greater (*p* < 0.05) in steers compared to heifers (78.54 ± 4.18 vs 67.80 ± 5.06 g/hd per day for steers and heifers, respectively). Feeder attendance varied from 0 to 77.80% of the total number of animals in the supplemented group (OH), and 3 animals never registered a visit to the feeder (Figure 3b). Average MLB intake among individual animals over the trial varied from 194.7 to 0 g/hd per day.

Herd averages of MLB intake, feeding rate, visit duration, and visit size were affected by feed availability and feed type (*p* < 0.05, Table 3). These feeding parameters were lower at high compared to low feed availability (*p* < 0.05, Table 3). In addition, MLB intake and feeding rate were higher while on OH compared to the rest of the feed types (*p* < 0.001, Table 3). Additionally, the intake of MLB was greater (*p* < 0.05) with low compared to high feed availability for all feed types except for OH (*p* > 0.05; Figure 4a). Sex did not affect MLB intake or feeding behaviour (*p* > 0.10). Feeding frequency and duration were affected by feed availability × feed type interaction (*p* < 0.05, Figure 4). Feeding frequency and feeding duration were lower at high compared to low feed availability while on P, OC, P+C, and LH (*p* < 0.05, Figure 4) but not while on OH (*p* > 0.10, Figure 4b,c).

### 3.3. Animal Performance

Overall, means LWC throughout the study were 435 ± 26 and 529 ± 25 g/day for NS and MLB groups (*p* < 0.05), respectively, representing an 18% increment with MLB. Liveweight change was not affected by sex or its interactions (*p* > 0.10). However, there was a significant interaction between MLB supplementation and time for LWC and LW (*p* < 0.05; Figure 5). Supplemented animals had greater LWC compared to NS animals only while on SG, P, and OH (*p* < 0.05, Figure 5). In addition, supplemented animals were heavier (+23.5 kg/hd) than NS animals at the end of the trial (*p* < 0.05; Figure 5). Liveweight was affected by sex being higher in steers compared to heifers (*p* < 0.05; data not shown).

## 4. Discussion

The objective of the present trial was to quantify the dynamic relationship between MLB intake; cattle growth rate (group averages); cattle feeding behavior; and forage type, quantity and quality by monitoring the effects of MLB supplementation on cattle LW under grazing conditions using EF and WOW. Continuous recording of LW and MLB intake enabled monitoring livestock throughout seasons and, therefore, to report the dynamic responses to type, quantity, and quality of feed available to them. One of the novel findings of the present study was that voluntary MLB intake and feeding behaviour are, on average, sensitive to changing types of forage, which can be useful to monitor and improve the nutritional management of animals. Such changes could go undetected or could be detected later without the combination of EF and WOW. To this effect, variations in MLB intake due to changes in forage quantity and quality could be detected by direct monitoring of block disappearance and MLB feeding behaviour (e.g., average daily feeding and duration). Moreover, results showed that LWC was highly variable across and within feed type because selective grazing decreases forage quantity and quality and that MLB supplementation was identified as one of the factors influencing LWC. In-paddock weighing combined with EF was shown to be useful to monitor the dynamic responses of cattle [1,3] in which actual consumption of the supplementary feed occurred only during certain times instead of the entire period. Remote monitoring of daily temporal variations of growth, liveweight, and supplement intake could help enhance the management of group-fed cattle.

Findings from the present trial suggest that feeding behaviour around MLB, including average feeding duration and frequency, reflect changes in MLB intake as a result of changes in feed quantity and quality. Therefore, remote monitoring of feeding behaviour around supplements could be useful to improve grazing management by reflecting such changes. To the best of our knowledge, this is the first study assessing the relationship between MLB feeding behaviour of individual animals and fed as a group and the quantity and quality of forage available measured over a long-term grazing study. Previous studies [14,15] also reported that the intake of molasses-based supplements can be influenced by forage availability but that feeding behaviour has not been adequately addressed in those studies [6]. Moreover, feeding rate was affected both across and within feed types, increasing at low feed availability even during short periods of rotational grazing (e.g., OC) or during hay supplementation. In addition, feeding rate increased when animals were consuming low-quality OH independently of forage availability. Previous studies reported that feeding rate is a robust indicator of feed characteristics, hunger, social competition, and feeding management [16]. Therefore, the greater MLB intake and feeding rate reported during low measurements in pastures and OH suggests animals were hungrier during such time periods due to low availability of nutrients.

However, feeding behaviour needs to be interpreted in relation to the type of supplement offered (MLB). The use of MLB aims to control intake and feeding rate through the block hardness, forcing animals to lick the supplement to consume it [5]. Thus, we expected that the ability of animals to increase feeding rate of MLB was limited compared to other supplements such as fodder or concentrates. However, licking rate per se was not measured and the results may also be affected by the composition of the block (i.e., % of CP, urea, and block hardness). Further studies are required to determine the ability of animals and the time required to change feeding rate according to changes in MLB composition. In line with this, Bowman et al. [6] concluded that the correlations between feeding duration, feeding frequency, and intake found with loose supplements could not be extrapolated to molasses supplements. Finally, feeding frequency and duration of MLB in the present study did not differ between high and low feed availability when animals were offered OH, suggesting that low-quality forages fed continuously could increase the basal attendance of animals to MLB. Thus, our results suggest that combining variations in feeding parameters of MLB could be useful, for example, as part of a procedure to automate the monitoring of forage quantity and quality.

An innovative aspect of this study is the simultaneous monitoring of LW and MLB intake, which has several implications for results interpretation. In this regard, MLB supplementation was one of the factors influencing LWC only during some periods of time. In addition, both MLB intake and the proportion of animals attending the feeders, which directly affects MLB intake, proved to be highly variable throughout the trial (within and across feed types). Variations in MLB could be a preliminary approach to future studies linking the increase of free-choice supplements with variations in LWC to anticipate further reductions, for example, by rotating animals to another paddock. Frequent and automatic data collection allows detecting periods when the effects of feed supplementation are more accentuated. This could be a relevant aspect as infrequent LW measures at group level (e.g., months) were probably masking supplementation periods with no responses in previous studies [6,8].

Positive effects of MLB supplementation on LWC were observed during periods when low- to medium-quality forages (SG, pastures, and OH) were offered but not with high-quality forages (LH and OC). Similar responses to molasses-based supplements were reported previously [6,10] and linked to improved forage intake and digestion [9]. Nevertheless, these results should be interpreted with caution as LWC responses and MLB intake between individual animals could largely differ, as we show in a companion paper (unpublished data) [17]. In addition to the nutritional benefits of supplementing MLB, ionophores included in the blocks (Lasalocid sodium) could have been effective to control common diseases of young cattle, such as coccidiosis, while improving feed efficiency [18,19]. The availability of minerals from MLB could also influence LWC of weaners as minerals are usually deficient in most forages in comparison to the high requirements of young growing cattle [20]. However, intake of minerals may be affected by the lack of uniform consumption by animals when it is offered as a free choice [21]. However, provision of minerals may be affected by the lack of uniform consumption by animals when it is offered as a free choice [21]. Therefore, both nutritional and health impacts of MLB could explain the better performance of MLB-supplemented animals, particularly at the beginning of the present trial as young cattle were introduced to new feed types after weaning. However, further research is needed to establish and quantify cause–effect relationships, e.g., the incidence of coccidiosis in grazing animals fed MLB with Lasalocid.

Remote LW monitoring of a group of animals proved to be highly relevant to identify average changes in daily LWC, even within the same feed type. In this regard, the use of in-paddock weighing may offer a novel platform to better understand the seasonal complexity (forage availability and quality) driving animal growth. It also reveals that managing nutrition to reduce LWC variability can be a challenging task, as animals could rapidly respond and adjust their LWC to factors modifying it. Additionally, monitoring the entire growth path of animals can be used to study the effects of previous growth on their current performance. For example, animals showing the same LWC may have different nutritional requirements and metabolic responses if they are experiencing an ascendant (e.g., OC and P+C) or descendant (e.g., LH) path of growth. Detecting these changes in LWC could also be crucial to understand periods of compensatory growth in cattle.

## 5. Conclusions

Average MLB intake of supplemented animals varied both between and within forage type. Such variations can be detected remotely in near real time using remote sensing, suggesting that continuous and simultaneous, automated monitoring of LWC and supplement intake could be useful to monitor changes in forage quantity and quality over time under grazing conditions.

## Figures and Tables

**Figure 1 animals-09-01129-f001:**
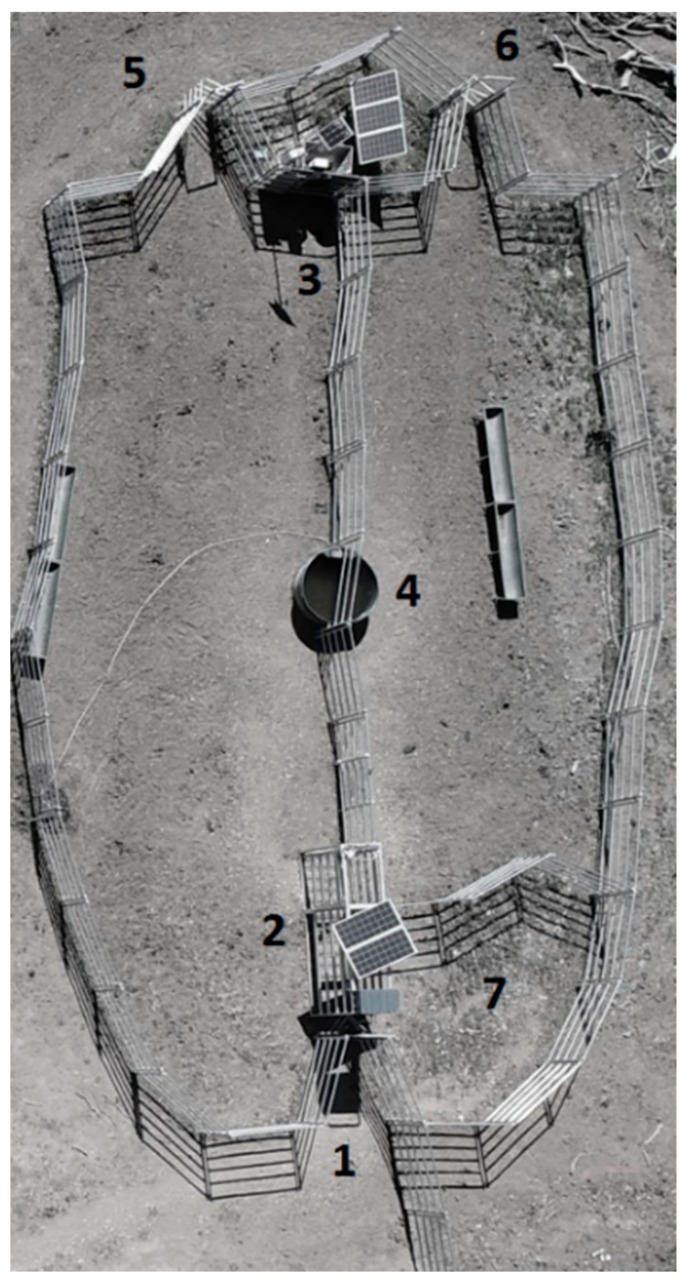
Image from a drone showing the water point enclosure with an in-paddock weighing station, auto-drafter gate, and electronic feeder to draft animals into a supplemented or unsupplemented group. Numbers refer to (1) enclosure entry; (2) locations of the weighing platform, auto-drafter gate, and solar powering equipment; (3) locations of the electronic feeder and solar powering equipment; (4) water point; (5) exit spear gate of supplemented animals; (6) exit spear gate of non-supplemented animals; and (7) internal enclosure to avoid electronic identification (EID) being recorded from animals inside the yard (not walking through the weighing platform).

**Figure 2 animals-09-01129-f002:**
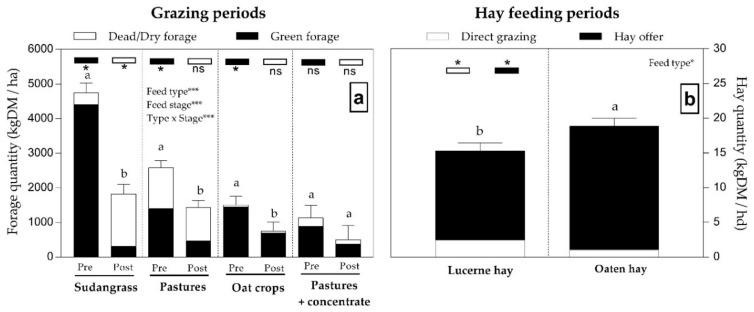
Forage quantity during periods of grazing (**a**) and hay feeding (**b**) of cattle supplemented with molasses-lick blocks or not: Grazing periods include forage quantity before (pre) and after (post) in each paddocks grazing period for each feed type. Hay-feeding periods include total feed availability (hay + forage from direct grazing) on days of hay delivery (Monday, Wednesday, and Friday). Different letters indicate significant differences (*p* < 0.05) between pre- and post-grazing (Figure 2a) and type of hay (Figure 2b). Asterisks indicate significant differences (* *p* < 0.05; *** *p* < 0.001).

**Figure 3 animals-09-01129-f003:**
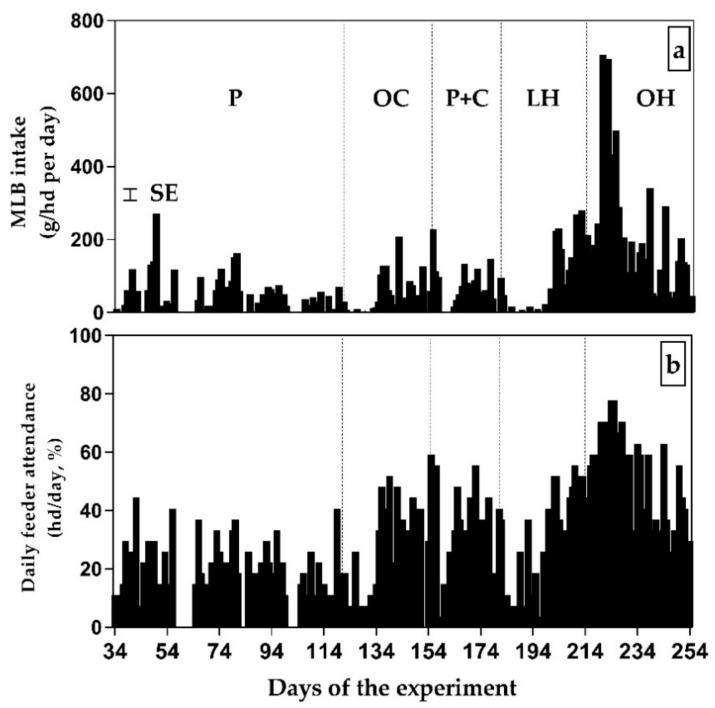
Daily molasses-lick-block (MLB) (**a**) intake (group average) and (**b**) feeder attendance (% of total herd) of cattle in a rotational grazing system: Animal attendance represents the number of animals consuming MLB within a day divided by the total number of animals of the herd. Feed types offered: P, Pastures; OC, Oat crops; P+C, Pastures with concentrate supplementation; LH, Lucerne hay; OH, Oaten hay.

**Figure 4 animals-09-01129-f004:**
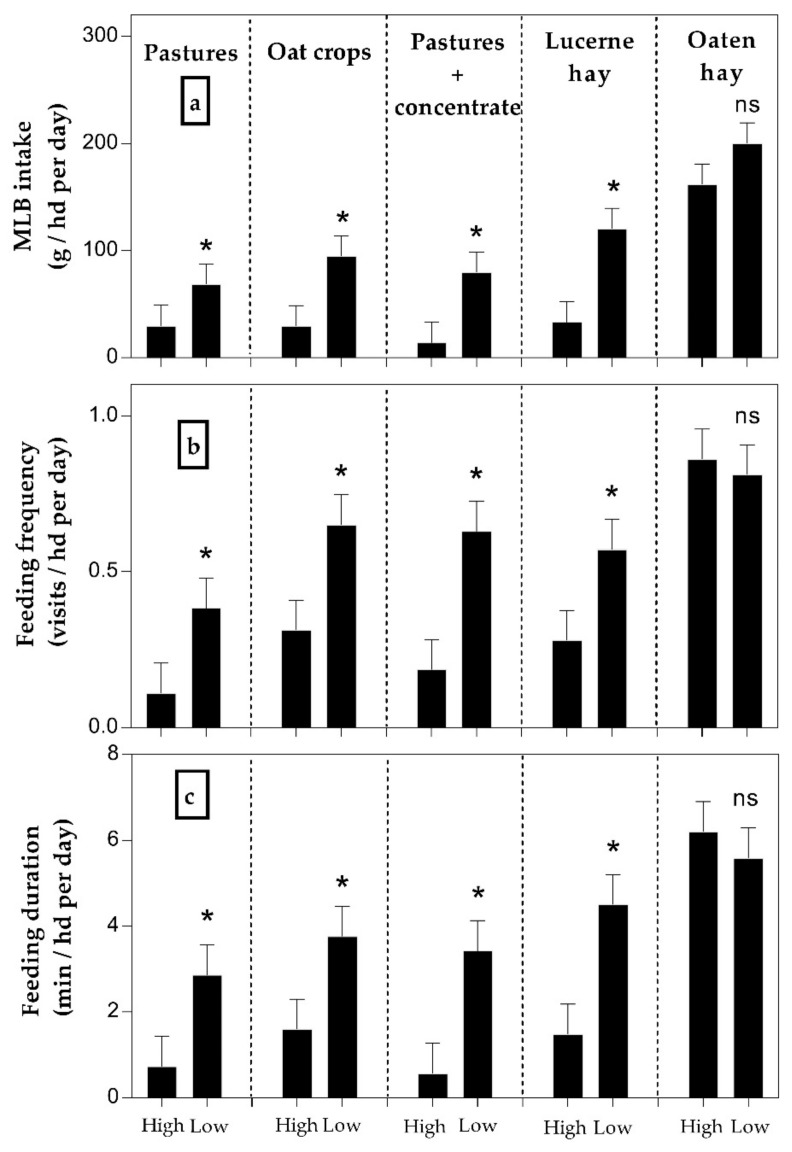
Molasses-lick-block intake (panel (**a**)) and feeding behaviour of cattle (Feeding frequency, panel (**b**); Feeding duration, panel (**c**)) while grazing five feed types at high and low forage availability. * Means differ within feed type (*p* < 0.05).

**Figure 5 animals-09-01129-f005:**
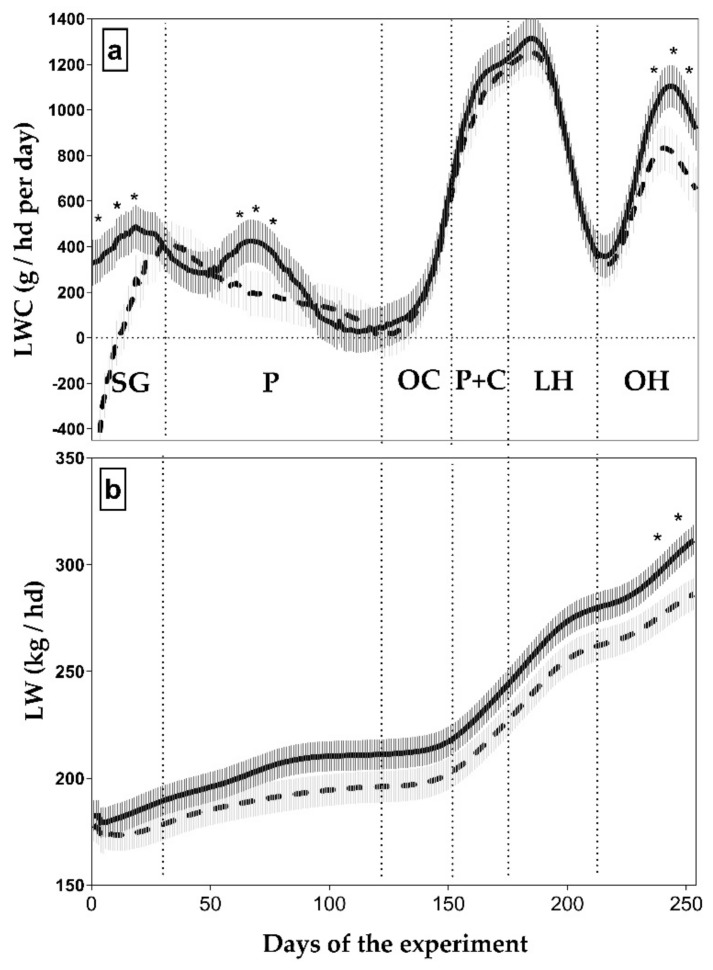
Daily liveweight change (LWC) (**a**) and liveweight (LW) (**b**) of cattle supplemented with molasses-lick blocks (MLB; solid line) or not supplemented (NS; broken line). Feed types offered: P, Pastures; OC, Oat crops; P+C, Pastures with concentrate supplementation; LH, Lucerne hay; OH, Oaten hay). * Means differ (* *p* < 0.05).

**Table 1 animals-09-01129-t001:** Chemical composition of grazed forages before (pre-) and after (post-) grazing by cattle offered supplementation with molasses-lick blocks or not.

Items	Sudangrass (SG)		Pastures (P)		Oat Crops (OC)		Pastures + Concentrate (P+C)		*p*-Values		
	Pre	Post	Pre	Post	Pre	Post	Pre	Post	Feed	Stage	F × S
CP (%)	12.6 ± 1.18	7.91 ± 1.182	8.04 ± 0.773	5.91 ± 0.892	14.5 ± 1.53	7.18 ± 1.531	14.5 ± 1.67	8.30 ± 1.674	<0.01	<0.01	0.17
NDF (%)	63.3 ± 1.56	66.6 ± 1.56	66.5 ± 1.18	71.5 ± 1.32	28.0 ± 2.54	41.1 ± 2.54	47.8 ± 2.54	56.4 ± 2.54	<0.01	<0.01	0.11
ADF (%)	33.5 ± 1.28	37.9 ± 1.28	38.2 ± 0.86	42.4 ± 0.96	14.6 ± 1.85	22.8 ± 1.85	25.5 ± 1.85	32.9 ± 1.85	<0.01	<0.01	0.29
DM (%)	19.9 ± 3.28 ^a^	23.7 ± 3.28 ^a^	33.29 ± 2.59 ^b^	46.2 ± 2.72 ^a^	19.4 ± 2.99 ^a^	22.8 ± 2.99 ^a^	29.4 ± 4.23 ^b^	41.8 ± 4.95 ^a^	<0.01	<0.01	0.04
OM (%)	89.7 ± 0.57 ^a^	91.4 ± 0.57 ^a^	91.7 ± 0.38 ^a^	92.6 ± 0.42 ^a^	94.0 ± 0.83 ^a^	90.6 ± 0.83 ^b^	89.8 ± 0.83 ^a^	90.4 ± 0.83 ^a^	0.03	0.87	<0.01
DMD (%)	59.2 ± 1.64	52.9 ± 1.64	55.7 ± 1.05	48.2 ± 1.35	92.8 ± 2.30	81.2 ± 2.30	77.2 ± 2.31	68.4 ± 2.31	<0.01	<0.01	0.58
DOMD (%)	57.0 ± 1.40	51.7 ± 1.40	54.0 ± 0.90	47.6 ± 1.15	85.4 ± 1.94	75.6 ± 1.94	72.1 ± 1.94	64.7 ± 1.94	<0.01	<0.01	0.57
Ash (%)	10.2 ± 0.57 ^a^	8.60 ± 0.571 ^b^	8.30 ± 0.394 ^a^	7.38 ± 0.427 ^b^	5.97 ± 0.838 ^b^	9.37 ± 0.832 ^a^	10.2 ± 0.83	9.5 ± 0.835	0.03	0.87	0.01

^a,b^ Means ± SE with different letters statistically differ (*p* < 0.05); CP = Crude Protein; NDF = Neutral Detergent Fiber; ADF = Acid Detergent Fiber; DM = Dry Matter; OM = Organic Matter; DMD = Dry Matter Digestibility; DOMD = Dry Organic Matter Digestibility.

**Table 2 animals-09-01129-t002:** Chemical composition of hay offered to cattle during supplementation with molasses-lick-blocks or not.

Items	Hay Offer		*p*-Values
	Lucerne (LH)	Oaten (OH)	Feed
CP (%)	21.9 ± 1.39	7.38 ± 1.141	<0.01
NDF (%)	32.6 ± 1.91	63.2 ± 2.34	<0.01
ADF (%)	24.3 ± 2.20	34.9 ± 1.79	<0.01
DM (%)	92.2 ± 2.53	93.5 ± 2.75	0.52
OM (%)	88.9 ± 0.94	93.3 ± 0.77	<0.01
DMD (%)	71.2 ± 3.07	58.0 ± 2.50	<0.01
DOMD (%)	67.1 ± 2.60	56.0 ± 2.13	<0.01
Ash (%)	11.1 ± 0.94	6.7 ± 0.772	<0.01

**Table 3 animals-09-01129-t003:** Feeding behaviour of cattle consuming molasses-lick-block supplement while grazing different types of forages. Means without a common letter differ (*p* < 0.05).

Items	Feed Availability				Feed Type						
	HIGH	LOW	SEM	*p*-Value	Pastures	Oat Crops	Pastures with Concentrate	Lucerne Hay	Oaten Hay	SEM	*p*-Value
MLB Intake (g/hd per day)	53.7	112.8	13.61	<0.01	49.4 ^b^	62.2 ^b^	46.9 ^b^	76.8 ^b^	181.3 ^a^	16.22	<0.001
Feeding Frequency (g/hd per day)	0.34	0.61	0.071	<0.001	0.24 ^b^	0.48 ^b^	0.40 ^b^	0.42 ^b^	0.83 ^a^	0.083	<0.001
Feeding Duration (g/hd per day)	2.12	4.02	0.512	<0.001	1.79 ^b^	2.68 ^b^	1.99 ^b^	2.99 ^b^	5.89 ^a^	0.592	<0.001
Feeding Rate (g/min)	19.6	24.0	1.87	<0.01	19.2 ^b^	19.7 ^b^	16.0 ^b^	23.3 ^b^	30.7 ^a^	2.67	<0.001
Visit Duration (min/visit)	4.92	6.18	0.353	<0.01	6.23 ^a^	5.03 ^a^	3.49 ^b^	6.16 ^a^	6.86 ^a^	0.554	<0.001
Visit Size (g/visit)	107.7	153.4	13.6	<0.001	135.6 ^c^	103.2 ^c^	57.8 ^d^	144.5 ^b^	211.5 ^a^	18.68	<0.001

^a,b,c,d^ Means with different letters statistically differ (*p* < 0.05).

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
