# Peer review of "Real-Time Monitoring of Self-Fed Supplement Intake, Feeding Behaviour, and Growth Rate as Affected by Forage Quantity and Quality of Rotationally Grazed Beef Cattle"

_animals, 2019, doi:10.3390/ani9121129_

Round 1
Reviewer 1 Report
General comments
I really enjoyed reading this paper which has the potential to have high impact in a very important area surrounding supplementary feed at grazing, especially within rangeland systems. Whilst the paper is well written my only major issue with the paper is the description of the design where the High feed availability and Low feed availability comparison and different paddock / feeding systems are not fully described as part of the design in section 2.1, which centres solely on the provision or not of the MLB. It therefore reads like an afterthought within the statistical analysis comparison - whereas this is the real novelty of the study and is the centre of the results in the abstract. The authors also do not describe how they dealt with the different breeds in the design and the statistical analysis of the study. Finally, the observations related to improved performance with the provision of MLB were attributed to a greater supply of macro-nutrients or ionophores from the MLB, although the provision of minerals could have equally resulted in the greater animal performance, especially on such low-quality feeds.
Specific comments
L3 Spelling of behaviour needs to be consistent throughout – check instructions to authors whether it is English or American English spelling that should be used.
L20 More description is required into the meaning of ‘certain periods’ as although a simple summary it needs to be meaningful.
L23 The abstract does not adequality describe the multifactorial design of the study – with aspects of High vs Low; MLB vs no MLB and SG vs P vs OC vs P+C to consider. Also provide more detail about the livestock mean age, weight and breed.
L69 Replace ‘due likely’ with ‘potentially due’
L72 Insert ‘:’ before introducing a list – here and throughout
L83-99 This section needs a lot more detail into the full multifactorial design of the study as mentioned above to include the comparison of High vs Low; MLB vs no MLB and SG vs P vs OC vs P+C and their interaction. Also, how were the two breeds allocated between the treatments and how was this effect dealt within the analysis. Be sure to provide all scientific names for species at first mention – i.e. cotton seed
L105 Replace ‘disappeared’ with ‘disappearing’
L108-117 How was access to the yard designed – was it driven entirely free-will driven by access to water or were animals corralled into the yard at certain times.
L150 Make it clear that both treatments (MLB vs no-MLB) co-grazed the paddocks with separation only occurring in the yard driven by access to water.
L154 Insert ‘(Medicago sativa)’ after ‘Lucerne’
L202-212 This section needs to be incorporate into a new 2.1 section to fully describe the design of the study. 2.5 should be about how the data was analysed based on the design described in section 2.1. The 2.5 section also needs to describe how breed effect was removed from the design.
L230 Replace ‘fit’ with ‘fitted’
L232 Here and throughout including tables – report means to 3s.f. and the variance associated with the mean to 1 d.p. greater than the mean i.e. 0.169 +/- 0.213X
L249 Were pre and post grazing samples taken before the animals first entered the paddocks and at the end of the grazing period as they left for the last time? If so how do they relate to the LOW and HIGH treatments?
Figure 2 – What is meant by early and late – how do these relate to LOW and HIGH treatments – be consistent with the terms and how what you are presenting relates to the design of the study.
Table 1 and 2 See comment on L232 and ASH is not an acronym – replace with Ash
L269 Replace ‘throught’ with ‘through’ for means see L232 comment.
L270 and L303 How did the results differ for the two breeds?
L322 Replace ‘sensible’ with ‘sensitive’
L345 For hunger was this driven by macro or micro nutrients? This needs to be considered and discussed.
L372 How did the animals LWC differ for the animals within the MLB treatment which did not visit the blocks – were they comparable to the non-MLB treatment? This is a good check to confirm the influence of the MLB on LWC.
L378 As mentioned above the authors need to consider and discuss the potential role for the provision of minerals.
L384 Why were the animals not assessed for coccidiosis, especially between MLB and control (no-MLB to determine whether this was a contributing factor to the results observed.
Author Response
General comments
a) Reviewer: I really enjoyed reading this paper which has the potential to have high impact in a very important area surrounding supplementary feed at grazing, especially within rangeland systems. Whilst the paper is well written my only major issue with the paper is the description of the design where the High feed availability and Low feed availability comparison and different paddock / feeding systems are not fully described as part of the design in section 2.1, which centres solely on the provision or not of the MLB. It therefore reads like an afterthought within the statistical analysis comparison - whereas this is the real novelty of the study and is the centre of the results in the abstract.
Authors: this comment was addressed in the new version of the manuscript. Section 2.1 was re-structured (line 97 to 116).
b) Reviewer: The authors also do not describe how they dealt with the different breeds in the design and the statistical analysis of the study.
Authors: the new version of the manuscript was corrected as it was a mistake, they were all crossbreds, not Charolais and Angus (line 82).
c) Reviewer: Finally, the observations related to improved performance with the provision of MLB were attributed to a greater supply of macro-nutrients or ionophores from the MLB, although the provision of minerals could have equally resulted in the greater animal performance, especially on such low-quality feeds.
Authors: this comment was addressed in the new version of the manuscript along with other comments related to mineral delivery using free-choice supplements as a vehicle of provision (line 380 to 384).
Specific comments
Reviewer: L3 / Spelling of behaviour needs to be consistent throughout – check instructions to authors whether it is English or American English spelling that should be used.
Authors: comment addressed in the new version of the manuscript.
Reviewer: L20 / More description is required into the meaning of ‘certain periods’ as although a simple summary it needs to be meaningful.
Authors: comment addressed in the new version of the manuscript (lines 20 to 22 and 33).
Reviewer: L23 / The abstract does not adequality describe the multifactorial design of the study – with aspects of High vs Low; MLB vs no MLB and SG vs P vs OC vs P+C to consider. Also provide more detail about the livestock mean age, weight and breed.
Authors: authors agree with this comment and we modified the abstract in order to include these details. However, the journal allows only 200 words and we had to prioritise contents (line 26 to 29).
Reviewer: L69 / Replace ‘due likely’ with ‘potentially due’
Authors: comment addressed in the new version of the manuscript.
Reviewer: L72 / Insert ‘:’ before introducing a list – here and throughout
Authors: comment addressed in the new version of the manuscript.
Reviewer: L83-99 / This section needs a lot more detail into the full multifactorial design of the study as mentioned above to include the comparison of High vs Low; MLB vs no MLB and SG vs P vs OC vs P+C and their interaction. Also, how were the two breeds allocated between the treatments and how was this effect dealt within the analysis. Be sure to provide all scientific names for species at first mention – i.e. cotton seed.
Authors: comment addressed in the new version of the manuscript and previously in the present document (crossbred).
Reviewer: L105 / Replace ‘disappeared’ with ‘disappearing’
Authors: comment addressed in the new version of the manuscript.
Reviewer: L108-117 / How was access to the yard designed – was it driven entirely free-will driven by access to water or were animals corralled into the yard at certain times.
Authors: the access to the yard was entirely free and driven by the access to the water as the only water point available which animals had access during the experiment (lines 104 to 106).
Reviewer: L150 / Make it clear that both treatments (MLB vs no-MLB) co-grazed the paddocks with separation only occurring in the yard driven by access to water.
Authors: comment addressed in the new version of the manuscript.
Reviewer: L154 / Insert ‘Medicago sativa’ after ‘Lucerne’.
Authors: comment addressed in the new version of the manuscript.
Reviewer: L202-212 / This section needs to be incorporate into a new 2.1 section to fully describe the design of the study. 2.5 should be about how the data was analysed based on the design described in section 2.1. The 2.5 section also needs to describe how breed effect was removed from the design.
Authors: this section was re-structured based on comments below. New section 2.1 has critical information to understand forages fed and how we considered scenarios of HIGH and LOW to analyse the intake of molasses-lick blocks (lines 82 to 116).
Reviewer: L230 / Replace ‘fit’ with ‘fitted’.
Authors: comment addressed in the new version of the manuscript.
Reviewer: L232 / Here and throughout including tables – report means to 3 s.f. and the variance associated with the mean to 1 d.p. greater than the mean i.e. 0.169 +/- 0.213X.
Authors: comment addressed in the new version of the manuscript (Tables).
Reviewer: L249 / Were pre and post grazing samples taken before the animals first entered the paddocks and at the end of the grazing period as they left for the last time? If so, how do they relate to the LOW and HIGH treatments?
Authors: Measures of forage quantity and quality were taken few hours before animals entered a fresh paddock or hay was offered, or a few hours after animals were pulled from a paddock. They relate to the HIGH and LOW (new version of the manuscript) for the analysis of MLB intake as pre- and post- grazing represent the closest approximation to the highest and lowest forage availability and quality measured. Sampling and measuring the forage when animals are already grazing a paddock (for example the second day of grazing) may led to underestimations of the forage they had available the day before and it is not possible to sample parts of the paddocks that were already depleted. Also, as feed types and grazing duration changed over time, we believe that measuring pre- and post- grazing condition (immediately closest to the days of MLB intake assessment) brought uniformity to this study. This is because instant stocking rate changed over the trial (growing animals) which would affect forage depletion rate and, as a result, the quantity and quality of the forage that we could measure with the animals already in the paddock. One of the aims behind this study was to understand variability in supplement intake of grazing animals which certainly does not allow to fully control feed allocation and quality as housed cattle do. This is one of the reasons why we included this tech platform to assess supplement intake and liveweight responses.
Reviewer: Figure 2 – What is meant by early and late – how do these relate to LOW and HIGH treatments – be consistent with the terms and how what you are presenting relates to the design of the study.
Authors: this section was re-structured in the new version of the manuscript and also related with the comment number 14.
Reviewer: Table 1 and 2 / See comment on L232 and ASH is not an acronym – replace with Ash.
Authors: comment addressed in the new version of the manuscript and changed in tables.
Reviewer: L269 / Replace ‘throught’ with ‘through’ for means see L232 comment.
Authors: comment addressed in the new version of the manuscript.
Reviewer: L270 and L303 / How did the results differ for the two breeds?
Authors: comment addressed in the new version of the manuscript, Crossbred weaners were used.
Reviewer: L322 / Replace ‘sensible’ with ‘sensitive’.
Authors: comment addressed in the new version of the manuscript.
Reviewer: L345 / For hunger was this driven by macro or micro nutrients? This needs to be considered and discussed.
Authors: comment addressed in the new version of the manuscript.
Reviewer: L372 / How did the animals LWC differ for the animals within the MLB treatment which did not visit the blocks – were they comparable to the non-MLB treatment? This is a good check to confirm the influence of the MLB on LWC.
Authors: we did explore individual differences in LWC among animals with contrasting MLB intake in another paper from this trial. A group of animals with lower MLB intake (e.g. those consuming less than the average) grew less than high consumers. It is certainly an interesting topic to discuss, however, MLB was just one among other factors affecting performance and out of the scope of the present study.
Reviewer: L378 / As mentioned above the authors need to consider and discuss the potential role for the provision of minerals.
Authors: comment addressed in the new version of the manuscript.
Reviewer: L384 / Why were the animals not assessed for coccidiosis, especially between MLB and control (no-MLB to determine whether this was a contributing factor to the results observed).
Authors: the present study focused on using in-paddock technologies in a long-term grazing experiment and describes growth trajectories of both groups of animals. Possible effects on coccidiosis was out of the scope of the present study but we did include speculations on the use of MLB as a vehicle to deliver medications. Once the performance and supplement intake are measured daily, there are many other factors to explore but we needed to limit our investigation.
Reviewer 2 Report
Well written paper. Use of automated recording devices is adding to our understanding of production. In my opinion, several issues (described below) must be addressed before this paper can be published.
Issue 1. Hypothesis. The data analysis does not address the hypothesis. To test the hypothesis, forage quality and quantity must be independent response variables and predictors (dependent variables) would be intake and feeding behavior. The data presented has intake and behavior as a response to forage type and forage mass.
Issue 2. Grazing management. It is not clear if MLB and No-MLB were grazed simultaneously on the same paddocks or each group was assigned separate paddocks. If they were on separate paddocks, can you distinguish whether or not responses were to MLB treatment or paddock without replicated paddocks as the experimental unit?
Issue 3. High and Low forage mass. Partitioning data into high and low forage mass in the stats analysis was described as taking the first 2 (or 1) days on a paddock and the last 2 (or 1) days on a paddock. I prefer the use of pre- and post- instead of high- and low- to describe the comparisons of intake and feeding behavior differences as you did in the first figures and tables. Forage mass was not managed consistently as a high or low treatment but were artifacts of when cattle were placed into and removed from a paddock. It is possible to conceive during a long grazing period with advancing maturity and changes in growing conditions for a given species that High toward the end of the study could be somewhat similar in quantity and quality to low at the beginning of the season.
Paddock management. Additional information is needed such as fertilization of paddocks for annuals establishment and clarification on whether sudans and oats were double cropped on the same paddocks.
Description of forage mass in figures and tables. Some describe FM as pre and post while others use high and low. Be consistent. Pre and Post preferred
It would be helpful to have a summary of mean and sd of paddock grazing days during each season.
Line by line suggestions
13-14 remove rate, simultaneously
42 is the / needed?
47 remove also
69 remove likely
90 Vegetal spelling?
153 check your grazing days – autumn ended on 122 but oats began on 121; oaten hay and Lucerne hay were both offered on day 220.
162 weighed prior to delivery
191 define NIRS
193 remove Chemical composition of
263 P value headers need to match model description in stats
276 Statement that Sex did not affect MLB intake. This contradicts line 267-271
316 I get lost in group averages. I thought the data was on individual animal weight and supplement intake. Was grouping daily average within animal? If animals were grouped, how were groups replicated and described in the stat methods?
362-366 were your number of visits and visit duration totals per day correlated with liveweight change?
Author Response
Comments and Suggestions for Authors
Well written paper. Use of automated recording devices is adding to our understanding of production. In my opinion, several issues (described below) must be addressed before this paper can be published.
Reviewer: Hypothesis. The data analysis does not address the hypothesis. To test the hypothesis, forage quality and quantity must be independent response variables and predictors (dependent variables) would be intake and feeding behavior. The data presented has intake and behavior as a response to forage type and forage mass.
Authors: the hypothesis of the new version of the manuscript was improved to reflect the data analysis performed (lines 73 to 74).
Reviewer: Issue 2. Grazing management. It is not clear if MLB and No-MLB were grazed simultaneously on the same paddocks or each group was assigned separate paddocks. If they were on separate paddocks, can you distinguish whether or not responses were to MLB treatment or paddock without replicated paddocks as the experimental unit?
Authors: animals from MLB and No-MLB grazed simultaneously on the same paddocks and were sorted by the auto-drafter at the entry of the yard. It was clarified in the new version of the manuscript with sentences, as for example, ‘It is important to stress that MLB-supplemented and NS animals co-grazed the paddocks and were sorted automatically into treatments at the entrance of the yard every time they wanted to access water’ (lines 104 to 106).
Reviewer: Issue 3. High and Low forage mass. Partitioning data into high and low forage mass in the stats analysis was described as taking the first 2 (or 1) days on a paddock and the last 2 (or 1) days on a paddock. I prefer the use of pre- and post- instead of high- and low- to describe the comparisons of intake and feeding behavior differences as you did in the first figures and tables. Forage mass was not managed consistently as a high or low treatment but were artifacts of when cattle were placed into and removed from a paddock. It is possible to conceive during a long grazing period with advancing maturity and changes in growing conditions for a given species that High toward the end of the study could be somewhat similar in quantity and quality to low at the beginning of the season.
Authors: terminology was extensively revised in the new version of the manuscript. We re-structured section 2.1 to make clear our experimental procedure. However, we decided to continue using HIGH and LOW as terminology. Our decision is based on avoiding confusions related to terminology associated with forage measures (pre- and post-) or when the intake of MLB and feeding behaviour of animals were measured (HIGH and LOW).
Paddock management. Additional information is needed such as fertilization of paddocks for annuals establishment and clarification on whether sudans and oats were double cropped on the same paddocks.
Authors: comment addressed in the new version of the manuscript (lines 184 to 185).
Description of forage mass in figures and tables. Some describe FM as pre and post while others use high and low. Be consistent. Pre and Post preferred.
Authors: comment addressed in the new version of the manuscript and based on general comment c).
It would be helpful to have a summary of Mean and SD of paddock grazing days during each season.
Authors: comment addressed in the new version of the manuscript.
Line by line suggestions
Reviewer: 13-14 remove rate, simultaneously
Authors: comment addressed in the new version of the manuscript.
Reviewer: 42 is the / needed?
Authors: comment addressed in the new version of the manuscript.
Reviewer: 47 remove also.
Authors: comment addressed in the new version of the manuscript.
Reviewer: 69 remove likely
Authors: comment addressed in the new version of the manuscript.
90 Vegetal spelling?
Authors: comment addressed in the new version of the manuscript.
Reviewer: 153 check your grazing days – autumn ended on 122 but oats began on 121; oaten hay and Lucerne hay were both offered on day 220.
Authors: comment addressed in the new version of the manuscript (lines 99 to 106).
Reviewer:162 weighed prior to delivery.
Authors: comment addressed in the new version of the manuscript.
Reviewer: 191 define NIRS.
Authors: comment addressed in the new version of the manuscript.
Reviewer:193 remove Chemical composition of.
Authors: comment addressed in the new version of the manuscript.
Reviewer: 263 P value headers need to match model description in stats.
Authors: comment addressed in the new version of the manuscript.
Reviewers: 276 Statement that Sex did not affect MLB intake. This contradicts line 267-271.
Authors: comment addressed in the new version of the manuscript.
Reviewers: 316 I get lost in group averages. I thought the data was on individual animal weight and supplement intake. Was grouping daily average within animal? If animals were grouped, how were groups replicated and described in the stat methods?
Authors: comment addressed in the new version of the manuscript. We used individual animal data (lines 318 to 322).
Reviewers: 362-366 were your number of visits and visit duration totals per day correlated with liveweight change?
Authors: MLB intake and feeding behaviour were correlated with liveweight change among individual animals subjected to MLB supplementation; however, it was out of the scope of the present manuscript and explored in other manuscript which is currently under review.